# Research on Dynamic Monitoring of Grain Filling Process of Winter Wheat from Time-Series Planet Imageries

**Xinxing Zhou, Yangyang Li, Yawei Sun, Yijun Su, Yimeng Li, Yuan Yi and Yaju Liu ***

Xuzhou Institute of Agricultural Sciences in Jiangsu Xuhuai District, Xuzhou 221131, China
* Correspondence: yajuliu@jaas.ac.cn; Tel.: +86-18796289532

**Abstract:** Remote sensing has been used as an important means of monitoring crop growth, especially for the monitoring of the formation of crop yield in the middle and late growth period. The information acquisition on the yield formation period of winter wheat is of great significance for winter wheat growth monitoring, yield estimation and scientific management. Hence, the main goal of this study was to verify the possibility of monitoring the grain-filling process of winter wheat and its in-field variability using an alternative non-destructive method based on orbital remote sensing. High-resolution satellite imageries (3 m) were obtained from the PlanetScope platform for three commercial winter wheat fields in Jiangsu Province, China during the reproductive stage of the winter wheat (185–215/193–223/194–224 days after sowing (DAS)). Based on the quantitative analysis of vegetation indices (VIs) obtained from high-resolution satellite imageries and three indicators of the winter wheat grain-filling process, linear, polynomial and logistic growth models were used to establish the relationship between VIs and the three indicators. The research showed a high Pearson correlation ($p < 0.001$) between winter wheat maturity and most VIs. In the overall model, the remote sensing inversion of the dry thousand-grain weight has the highest accuracy and its $R^2$ reaches more than 0.8, which is followed by fresh thousand-grain weight and water content, the accuracies of which are also considerable. The results indicated a great potential to use high-resolution satellite imageries to monitor winter wheat maturity variability in fields and subfields. In addition, the proposed method contributes to monitoring the dynamic spatio-temporality of the grain-filling progression, allowing for more accurate management strategies in regard to winter wheat.

**Keywords:** time-series planet imageries; winter wheat; grain filling; remote sensing; vegetation indices

## 1. Introduction

Wheat (*Triticum aestivum* L.) is one of the major staple crops globally and a major source of calories and proteins in Northern China [1,2]. As the largest food consumer in the world, China is currently facing multiple pressures, such as the reduction in cultivated land and environmental degradation. Therefore, only by selecting improved varieties and high-yield cultivation techniques can we break through the yield limit and achieve the goal of increasing production. The current high-efficiency and high-quality production of wheat is of great significance to solving the problem of food safety and stabilizing social and economic development [3–5]. The grain-filling period of wheat is a critical period for the formation of wheat yield and quality. Scientific, accurate and rapid acquisition of relevant indicators during the wheat grain-filling period and timely monitoring of the wheat grain-filling process can provide a reliable reference for field management [6–8].

The current research on the wheat grain-filling process mainly focuses on the simulation of the growth process of dry grain weight under different varieties or environments, as well as the further estimation of parameters such as the grain-filling rate and grain-filling duration, and conducting correlation analyses with varieties or environmental factors to explore the wheat grain-filling characteristics under different varieties or different environmental factors [9–13]. Usually, during the whole period from the start of grain filling

to maturity, it is necessary to continuously and repeatedly select representative wheat ear samples in the field for destructive sampling, then carry out further manual threshing to measure the fresh weight and drying to measure the dry weight of the grain, and finally, perform further statistical analysis. This process is not only time consuming and labor intensive, but also requires destructive sampling during multiple observations, resulting in limited sample data. In addition, the data obtained are often in the form of points, which are different from the actual situation in the field; thus, it is impossible to monitor the wheat grain-filling process in a visual, accurate and dynamic way.

To improve the accuracy of monitoring the grain-filling progress, several researchers have been looking for alternative methods, mainly by using digital image analysis [14,15]. This technique reduces human errors due to an individual's subjectivity in judging wheat maturity, but still presents other challenges. Digital image evaluation is limited by its monitoring range and spectral band information. The standardized acquisition of photos is also very important as, for instance, the amount of light when the photos are taken can seriously affect the results [16,17]. Classifying the wheat maturity using an image method solves the subjectivity of human viewing, but can add other sources of error in the classification process and does not represent the variability within a field.

An alternative method consists of using remote sensing to map wheat maturity in a field and creating zonal management strategies that can account for maturity variability. Due to the macroscopic, objective, timely and economical characteristics of remote sensing technology, along with the development of information technology and the modernization requirements of precision agriculture and smart agriculture, spatial information technology represented by remote sensing technology has been widely used in the field of agriculture [18–22]. Remote sensing technology has been widely studied in the monitoring of crop maturity [23–25], and most of the studies were based on the following two methods: one of them is to use time-series remote sensing data to monitor the whole growth period of crops and to analyze the changes of characteristic parameters at the end of crop growth to determine the maturity period of crops; another way is to use remote sensing data to quantify physiological and biochemical parameters related to crop maturity characteristics, such as the leaf area index (LAI), leaf chlorophyll content (LCC), etc., and then achieve a crop maturity assessment [26–28].

At present, there are a lot of studies using remote sensing technology to monitor the growth of winter wheat and predict its yield and quality, but there are only a few studies using remote sensing technology, especially satellite imageries, to monitor the whole grain-filling process of winter wheat [29–31]. Remote sensing has been reported as a potential tool to monitor winter wheat maturity, but because of the lack of additional studies, results reported thus far do not agree on the most appropriate vegetation indices to use, justifying the need for new studies, especially under different varieties [32]. Therefore, based on the fact that winter wheat maturity monitoring methods are still highly subjective and do not consider the variability of plants in the field, utilizing satellite imageries can help to solve this issue.

In view of the demand characteristics of large-scale monitoring in agriculture, the current remote sensing imageries used in crop monitoring were mainly originated via MODIS, NOAA/AVHRR, etc. [33–35]. In recent years, with the in-depth exploration of the potential of Sentinel series data and Landsat series data, the application of medium- and high-resolution multispectral satellite remote sensing imageries in precision agriculture has been brilliant [31,36,37]. However, these imageries have a relatively low spatial resolution and are difficult to apply to the high-precision remote sensing monitoring of winter wheat at the small field level. On the other hand, considering the characteristics of satellite imageries and the influence of weather factors, the revisit cycle of these medium- and high-resolution imageries is several days, which is relatively long, and it is difficult to obtain high-quality data in time. This limits the continuous monitoring of crops, making it difficult for crops to be monitored for specific indicators during critical growing periods. PlanetScope is the only remote sensing satellite system with global high-resolution, high-frequency, full

coverage in the world. Currently, more than 100 satellites have been launched. They form a satellite constellation to ensure that imageries with a resolution of 3–5 m are obtained every day in any area. Since 2021, the newly launched SuperDoves have a fifth spectral band (red-edge), which is valuable for plant health assessments including NDRE. Time-series satellite imageries with high-spatial and -temporal resolution can better monitor the growth and health of crops and improve farming efficiency and monitoring accuracy in fields and subfields [38–40].

The main objectives of this research using high-temporal/spatial-resolution imageries were: (I) to provide evidence of the potential use of satellite imageries in monitoring the variability of the winter wheat grain-filling process and identify potential vegetation indices to monitor the maturity variability in fields and subfields; and (II) to propose a new methodology to monitor winter wheat maturity using vegetation indices to assist winter wheat growers in improving management strategies for their fields. It is hypothesized that vegetation indices derived from the satellite imageries can be used to monitor variability in the winter wheat grain-filling process, thus improving the precision of field management.

## 2. Materials and Methods

### 2.1. Study Region Experimental Design

The experiments were conducted in a modern agricultural demonstration base in Xuzhou City, Jiangsu Province, China. As a large-scale, high-standard experimental field integrating scientific research and production, the base is rich in plant types to meet the needs of modern agricultural production (Figure 1). Rice–wheat rotation is a common farming method in Jiangsu Province, China, in which the crop rotation of the Xumai 44 and Xumai 43 fields was rice. Additionally, the crop rotation of the Xumai 33 field was sweet potato. The three winter wheat varieties are widely grown in northern Jiangsu. After the mechanized harvest of autumn grain crops, rice, sweet potato and soybeans are removed and winter wheat is planted, usually from October 15th until the start of November, at the demonstration base. The coldest months in this region are December and January (winter). These months offer favorable climatic conditions to winter wheat development, since winter wheat plants need vernalization, i.e., they need a period of low temperature before they can enter the reproductive growth phase from the vegetative growth phase (Figure 2). Xumai 44, Xumai 43 and Xumai 33 were sown on 21 October 2021, 20 October 2021 and 29 October 2021, respectively. The irrigation and fertilization management practices were the same for all studied fields. These cultivars reach the grain-filling period approximately 180 days after sowing (DAS).

### 2.2. Field Data Acquisition

The field data were built from information obtained in each of the georeferenced sampling plots, which were distributed in a regular grid for each study field. There were 16 long-term observation points for each cultivar field, totaling 48 in 3 fields (Figure 3). During the grain-filling period, the five-point sampling method was used for data acquisition at each sampling plot. As shown in Figure 3, the size of each sampling plot was controlled within 2 m under the measurement of tape. Then, a rectangular frame with a fixed size of 0.1 m$^2$ was used, and 5 small points were fixed at the 4 corners and in the middle of the sampling plot. In each 0.1m$^2$ small point, 5 representative winter wheat ears were randomly selected and put in the sampling bag. The winter wheat ears of the five plots were brought back to the laboratory, and then were shelled manually. After weighing the fresh weight of the winter wheat grains, the grains were put into an oven and were dried at a constant temperature of 80 °C for 48 h to obtain the dry weight. After obtaining the fresh grain weight and dry grain weight, the water content data of the grain could be obtained through the ratio relationship. After averaging the field data of these 5 points, the average value was the value of the sampling site, which comprised the dry thousand-grain weight, fresh thousand-grain weight and water content.

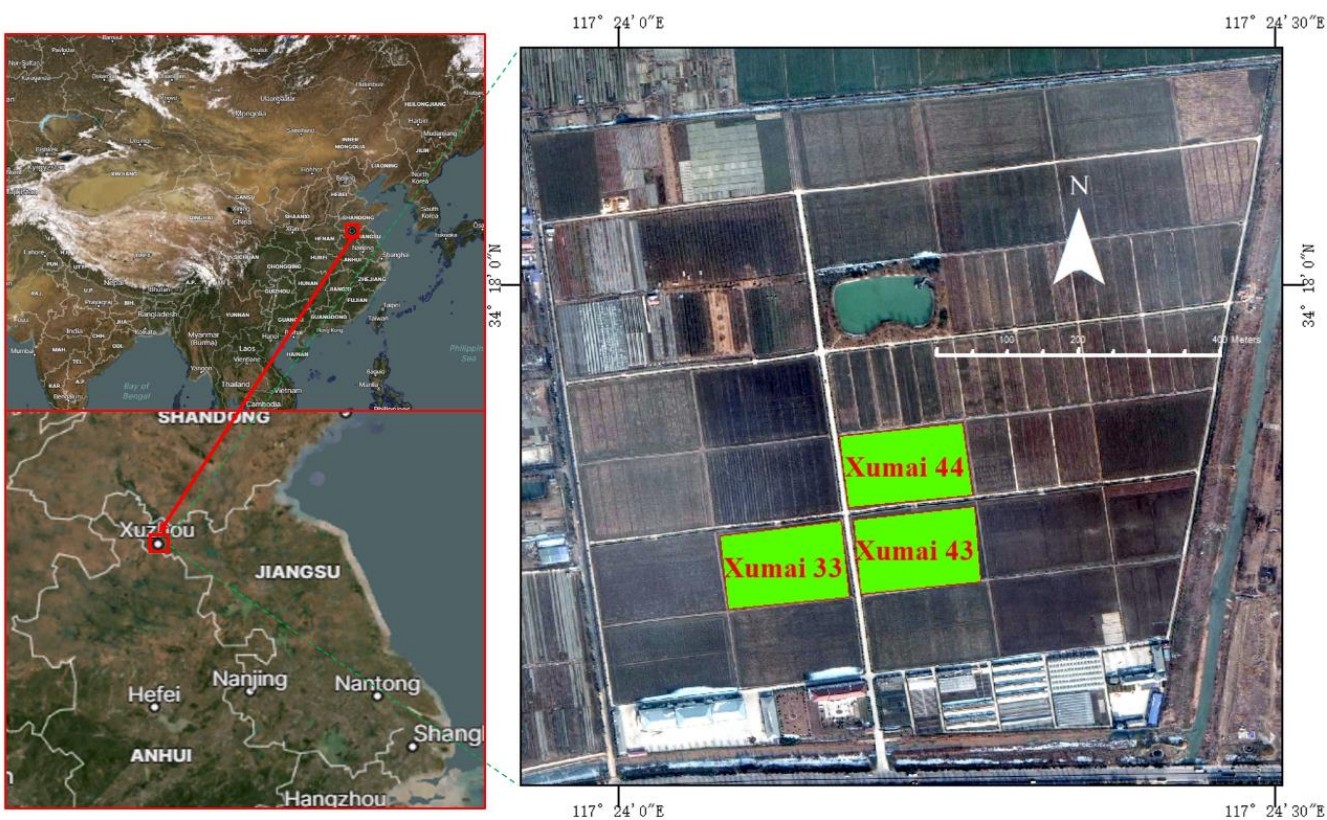

**Figure 1.** Location of the study area.

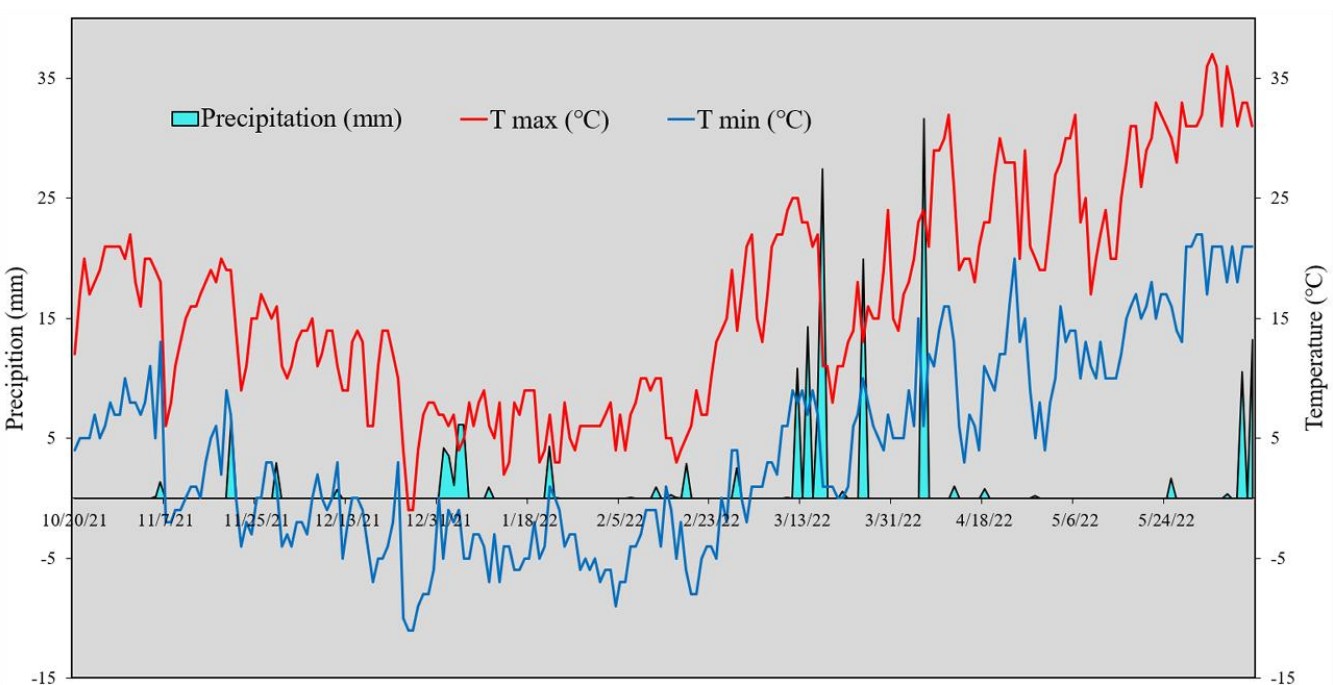

**Figure 2.** Daily precipitation, and maximum and minimum temperature for study area during 2021/2022 growing season.

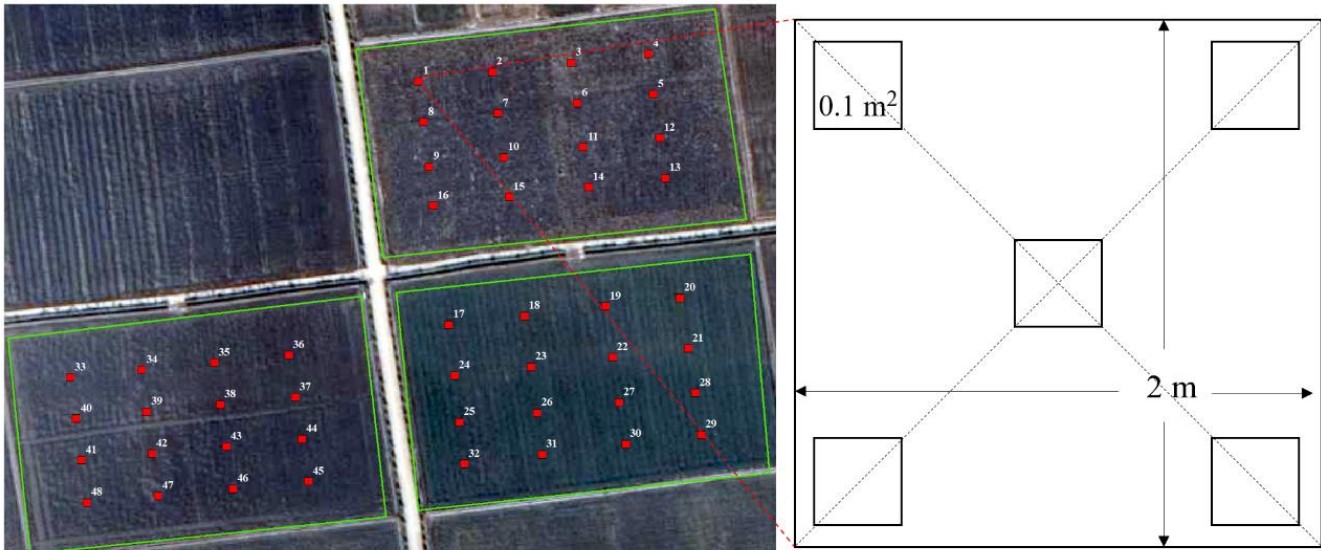

**Figure 3.** Location of georeferenced sampling plots in the field and five-point sampling method.

*2.3. Remote Sensing Imagery Data*

　　The imageries used were taken from the PlanetScope CubeSat platform, which has satellites characterized as 3U CubeSats (10 × 10 × 30 cm) with a mass of approximately 4 kg. PlanetScope CubeSats consist of an optical sensor satellite constellation with high resolution. Currently, there are approximately 170 satellites in solar orbit, which can collect imageries daily from anywhere on the globe with a resolution of 3–5 m. With the official launch of a new generation of PlanetScope eight-band products, on the one hand, Planet has further optimized and improved the key processing of imageries, which is reflected in reducing the number of pixels with poor quality, and improving the registration accuracy between bands. On the other hand, in addition to the existing red (650–680 nm), green (513–549 nm), blue (465–515 nm) and near-infrared (845–885 nm) bands, the coastal blue (431–452 nm), red-edge (697–713 nm), yellow (600–620 nm) and second green (513–549 nm) bands have been added, which are widely used in coastal zones, surface type identification, crop growth assessment, yield estimation, environmental monitoring and so on [41,42]. Five cloud-free satellite imageries were obtained of the modern agricultural demonstration base throughout the grain-filling stage from the Planet Labs platform. All satellite imageries were downloaded as Surface Reflectance (SR) products from www.planet.com accessed on 17 September 2022. The characteristics of the imageries are shown in Table 1.

**Table 1.** Characteristics of PlanetScope satellite imageries (3 m resolution) on different dates of the grain-filling stage.

| Date | DAS | | | | Time (UTC) | Sun Elevation | Sun Azimuth |
|------|-----|---|---|---|------------|---------------|-------------|
| | Period | Xumai 44 | Xumai 43 | Xumai 33 | | | |
| 1 May 2022 | A | 193 | 194 | 185 | 13:57 | 55.4° | 115.6° |
| 6 May 2022 | B | 198 | 199 | 190 | 14:44 | 64.2° | 129.1° |
| 15 May 2022 | C | 207 | 208 | 199 | 14:31 | 63.7° | 119.9° |
| 23 May 2022 | D | 215 | 216 | 207 | 14:33 | 64.7° | 116.8° |
| 15 June 2022 | E | 223 | 224 | 215 | 13:58 | 58.9 | 104.1° |

*2.4. Vegetation Indices (VIs)*

　　The vegetation indices are an important indicator to measure the growth of crops, monitor the crop development process, and estimate the crop yield potential. In combination with the physical significance of spectral indices, the selection of VIs was based on the spectral characteristics of crops and the available literature from China and abroad.

In the research of the monitoring of the crop growth process and yield formation, the normalized differential vegetation index (NDVI), green normalized differential vegetation index (GNDVI), plant senescence reflectance index (PSRI), normalized difference red-edge index (NDRE), normalized difference vegetation index of red-edge/red (NDVI-rededge) and nonlinear vegetation index (NLI), which are six VIs based on visible light, red-edge and near-infrared wavebands, were widely used. ArcMap 10.7 software (ESRI Co., Redlands, CA, USA) was used to extract reflectance data from all bands, and SPSS 25 (IBM Co., Foster City, CA, USA) software was used to calculate the six VIs.

All VI values were used to correlate with the dry thousand-grain weight, fresh thousand-grain weight and water content at each sampling point, and the linear model, polynomial model and logistic growth model were created to validate the potential of these VIs in predicting winter wheat maturity variability. These indices were selected due to their wide application in monitoring different physiological and agronomical parameters in various crops (Table 2).

**Table 2.** Vegetation indices used to monitor the grain-filling process of winter wheat.

| VI | Formula | References |
|---|---|---|
| NDVI | (NIR-Red)/(NIR+Red) | [33] |
| GNDVI | (NIR-Green)/(NIR+Green) | [43] |
| PSRI | (Red-Blue)/NIR | [44] |
| NDRE | (NIR-Rededge)/(NIR+Rededge) | [45] |
| NDVI-rededge | (Rededge-Red)/(Rededge+Red) | [46] |
| NLI | (NIR2-Red)/(NIR2+Red) | [47] |

*2.5. Data Analysis*

The Pearson correlation ($p < 0.001$) was used to verify the correlation between the VIs and winter wheat maturity during the 5 consecutive periods and to verify the overall correlation for each field. The development speed of crop yield formation in different stages is generally different, and usually, a growth curve model can be used to achieve better fitting and prediction effects. In order to enhance the stability of the monitoring model, the numerical samples of the wheat grain-filling process indicators in five periods were randomly divided into model development and model validation according to a ratio of 3:1. On the basis of the correlation analysis, the logistic model was selected to adjust the VI value to the grain-filling process according to the winter wheat growth development (Equation (1)), and linear and polynomial relationships were also established for each significantly correlated vegetation index for comparison.

$$W = W_f / [1 + exp^{-k(g-M)}] \tag{1}$$

where $W$ represents the response surface (wheat grain-filling process indicators), $W_f$ is the maximum value (upper asymptote), $k$ is the relative growth rate in $M$, $g$ is the variable (VIs), and $M$ is the VIs at which the growth rate is maximized.

Using the samples of the model development and model validation, the model was evaluated by plotting the 1:1 relationship graph between the predicted and measured values of the indicators of the wheat grain-filling process. The root-mean-square error (RMSE) and $R^2$ were used to measure the accuracy and precision of each monitoring model. Monitoring models with a low RMSE and high $R^2$ indicate that a vegetation index has high accuracy and precision in estimating the wheat grain-filling process. The $R^2$ and RMSE were calculated using Equations (2) and (3):

$$R^2 = 1 - \Sigma_{i=1}^{n}(y_i - \hat{y}_i)^2 / \Sigma_{i=1}^{n}(y_i - \overline{y})^2 \tag{2}$$

$$RMSE = \sqrt{\Sigma_{i=1}^{n}(y_j - \hat{y}_i)^2 / n} \tag{3}$$

where $y_i$ and $\hat{y}_i$ represent measured values and predicted values of indicators of the wheat grain-filling process, respectively, $\overline{y}$ is the average value of indicators of the wheat grain-filling process, and $n$ is the number of samples.

Vegetation indices with the highest performance (RMSE and $R^2$) in the overall analysis were selected to represent the variability in indicators of the wheat grain-filling process. The best performing model equations were used to estimate indicators of the wheat grain-filling process from VIs to create wheat maturity maps. For that, VI values for each pixel in the downloaded imageries were inputted into the model equation and the wheat grain-filling process was estimated for each pixel at each sample date.

## 3. Results

### 3.1. Quantitative Analysis between VIs and Grain Character of Winter Wheat

The quantitative analysis of samples and VIs of three varieties in different time periods and overall time illustrated that there were significant or extremely significant relationships between the indicators of the wheat grain-filling process and most VIs in some periods, and the significant relationship was stronger for the overall time (Figure 4). In terms of wheat dry thousand-grain weight and fresh thousand-grain weight, there were many negative correlations between VIs and their indicators in each period. In the relationship between VIs and dry thousand-grain weight, the correlation gradually strengthened from DAS A to DAS D, and decreased in DAS E. The overall analysis of the data indicated that there were VIs that had the most significant correlation with the dry thousand-grain weight in three varieties (NDRE, r = −0.83; PSRI, r = 0.82). In the relationship between VIs and fresh thousand-grain weight, the correlation showed, in general, a progressively stronger trend from DAS A to DAS E. The overall analysis of the data indicated that there were VIs that had the most significant correlation with the fresh thousand-grain weight in three varieties (NDRE, r = −0.76). In terms of wheat grain water content, there were many positive correlations between VIs and their indicators in each period, except the PSRI. In DAS E, the correlation between VIs and water content reached the maximum (PSRI, r = −0.81). In the overall analysis of the data, the correlation between the PSRI, NDRE and VIs reached the largest absolute value of 0.76, followed by the NDVI and NLI, both of which were 0.75. These results indicated that the VIs could reflect the grain-filling process of winter wheat well in the time span.

### 3.2. Model Construction for Monitoring the Grain-Filling Process of Winter Wheat

Based on the principle of the strong correlation and the above analysis results, the VIs with a better performance were selected to construct linear, polynomial and logistic growth models. Additionally, for three different winter wheat varieties, the VI-based monitoring modeling of the grain-filling process was carried out, i.e., the prediction models were built using data from three varieties, individually and combined (overall). The VI independent variables corresponding to different function models were screened out via the best $R^2$ performance of the model development and the model validation (Table 3).

In the monitoring model of the overall data, the best performing VIs were mainly the NDRE and PSRI. When modeling different varieties, it could be found that the best VI for the three indicators of Xumai 44 and Xumai 43 was the NDRE, and the best VIs of Xumai 33 were the NDRE, NLI, NDVI and PSRI. In addition, during the screening process of functional models, it was found that the logistic growth model did not perform the best on all varieties and all indicators. The model of the overall data showed that the prediction accuracy of the logistic growth model was better than that of the linear and polynomial models in both dry and fresh thousand-grain weights, and was slightly worse than the polynomial model in terms of water content. In the separate modeling of the three varieties, it could be found that in terms of the monitoring of dry and fresh thousand-grain weight, all varieties performed the best on the logistic growth model. In the monitoring model of water content, except for Xumai 33, which performed slightly better on the logistic growth model, all other varieties performed the best on the polynomial model.

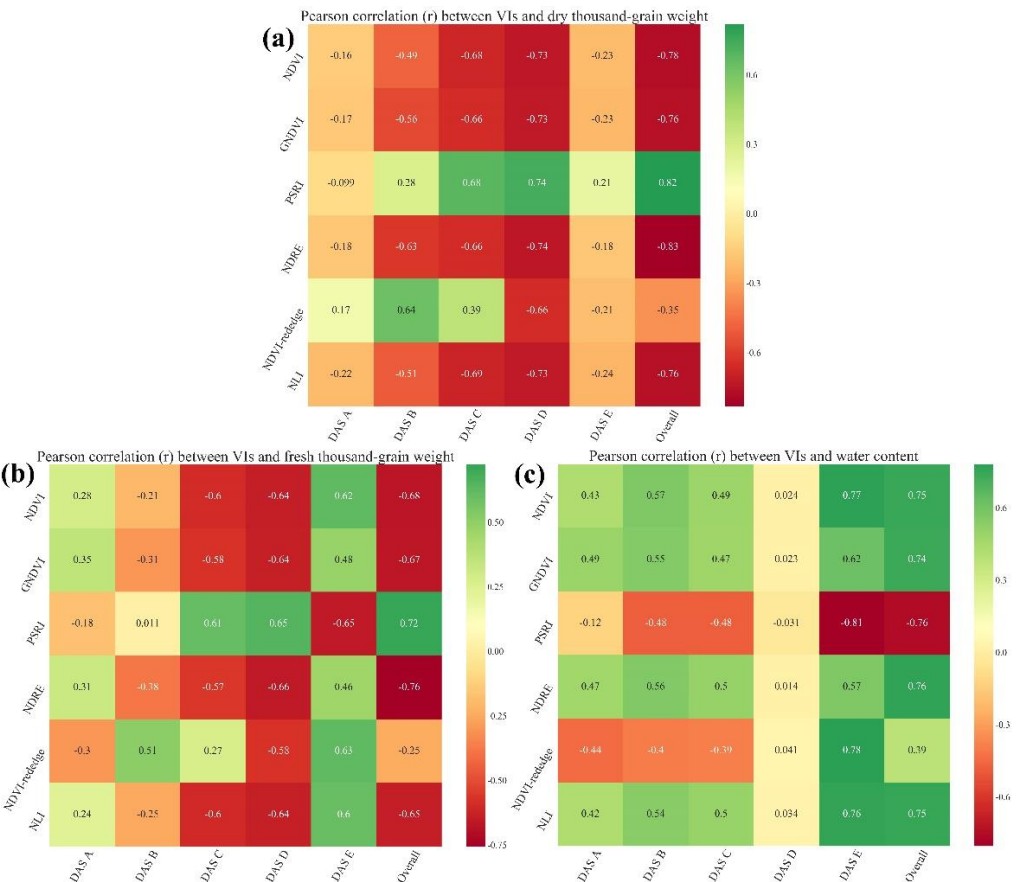

**Figure 4.** Pearson correlation (r) between vegetation indices (VIs) and three indicators ((**a**–**c**) represent dry thousand-grain weight, fresh thousand-grain weight and water content, respectively) of wheat grain-filling process by days after sowing and overall. A–E represent five sampling periods.

Overall correlations were strong despite the decrease in accuracy of the monitoring models ($R^2$, RMSE) when compared to the individual variety models (Figure 5). The VIs in the model showed a positive correlation trend with the indicators of the winter wheat grain-filling process on the whole, and the fitting effect performed well, indicating the potential to use a single model to estimate wheat maturity independently of the varieties.

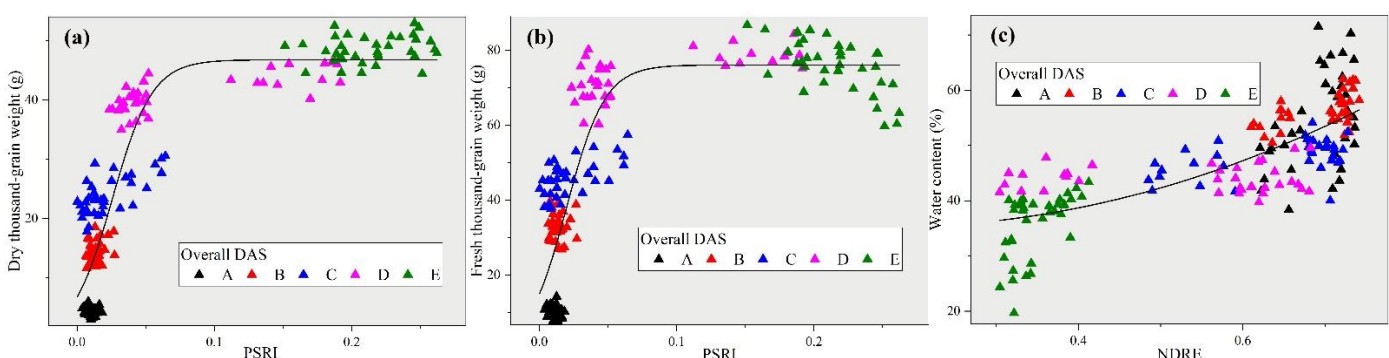

**Figure 5.** Overall models to monitor dry thousand-grain weight, fresh thousand-grain weight and water content using PSRI (**a**,**b**) and NDRE (**c**). A–E represent five sampling periods.

**Table 3.** Comparison of monitoring models for the grain-filling process of winter wheat under different indicators and functions.

| Variety | Indicators | Model | VI | Number of Samples | | $R^2$ | | RMSE | | Unit |
|---|---|---|---|---|---|---|---|---|---|---|
| | | | | Model Development | Model Validation | Model Development | Model Validation | Model Validation | Model Development | |
| Overall | Dry thousand-seed weight | Linear | NDRE | | | 0.69 | 0.72 | 9.1717 | 9.0301 | |
| | | Polynomial | PSRI | 180 | 60 | 0.76 | 0.75 | 8.0978 | 8.5644 | |
| | | Logistic | PSRI | | | 0.82 | 0.83 | 7.0945 | 7.0808 | g |
| | Fresh thousand-seed weight | Linear | NDRE | | | 0.57 | 0.64 | 16.6318 | 15.8382 | |
| | | Polynomial | PSRI | 180 | 60 | 0.69 | 0.70 | 14.5136 | 14.1478 | |
| | | Logistic | PSRI | | | 0.73 | 0.77 | 13.1672 | 12.7791 | |
| | Water content | Linear | | | | 0.58 | 0.56 | 5.25 | 5.37 | |
| | | Polynomial | NDRE | 180 | 60 | 0.60 | 0.59 | 5.18 | 5.23 | % |
| | | Logistic | | | | 0.59 | 0.58 | 5.21 | 5.31 | |
| Xumai 44 | Dry thousand-seed weight | Linear | | | | 0.75 | 0.71 | 8.1252 | 8.5266 | |
| | | Polynomial | NDRE | 60 | 20 | 0.90 | 0.90 | 5.0345 | 4.9375 | |
| | | Logistic | | | | 0.91 | 0.92 | 4.8674 | 4.5659 | g |
| | Fresh thousand-seed weight | Linear | | | | 0.66 | 0.62 | 14.7975 | 14.9758 | |
| | | Polynomial | NDRE | 60 | 20 | 0.85 | 0.83 | 9.9418 | 9.9574 | |
| | | Logistic | | | | 0.86 | 0.86 | 9.6176 | 9.9292 | |
| | Water content | Linear | | | | 0.57 | 0.55 | 5.48 | 5.88 | |
| | | Polynomial | NDRE | 60 | 20 | 0.65 | 0.71 | 4.90 | 3.91 | % |
| | | Logistic | | | | 0.59 | 0.57 | 5.36 | 5.73 | |
| Xumai 43 | Dry thousand-seed weight | Linear | | | | 0.66 | 0.69 | 9.5365 | 9.4525 | |
| | | Polynomial | NDRE | 60 | 20 | 0.86 | 0.86 | 6.0708 | 6.3522 | |
| | | Logistic | | | | 0.87 | 0.88 | 5.9170 | 5.7755 | g |
| | Fresh thousand-seed weight | Linear | | | | 0.56 | 0.65 | 16.9642 | 15.7713 | |
| | | Polynomial | NDRE | 60 | 20 | 0.77 | 0.76 | 12.8213 | 12.9819 | |
| | | Logistic | | | | 0.80 | 0.79 | 11.4768 | 12.3410 | |
| | Water content | Linear | | | | 0.50 | 0.38 | 6.05 | 6.39 | |
| | | Polynomial | NDRE | 60 | 20 | 0.66 | 0.76 | 5.00 | 3.99 | % |
| | | Logistic | | | | 0.52 | 0.41 | 5.93 | 6.25 | |
| Xumai 33 | Dry thousand-seed weight | Linear | | | | 0.92 | 0.88 | 4.6962 | 6.3441 | |
| | | Polynomial | NDRE | 60 | 20 | 0.93 | 0.88 | 4.4737 | 6.2272 | |
| | | Logistic | | | | 0.93 | 0.90 | 4.5068 | 6.0222 | g |
| | Fresh thousand-seed weight | Linear | NDRE | | | 0.81 | 0.76 | 11.7217 | 13.3073 | |
| | | Polynomial | NDVI | 60 | 20 | 0.86 | 0.80 | 9.4637 | 12.7224 | |
| | | Logistic | PSRI | | | 0.85 | 0.83 | 9.8304 | 11.1284 | |
| | Water content | Linear | NLI | | | 0.79 | 0.77 | 4.28 | 4.60 | |
| | | Polynomial | NDVI | 60 | 20 | 0.80 | 0.72 | 4.15 | 5.81 | % |
| | | Logistic | NLI | | | 0.80 | 0.73 | 4.12 | 5.67 | |

In the single-cultivar model, the VIs of Xumai 44 and Xumai 43 remained relatively consistent in terms of dry and fresh thousand-grain weight and water content, and were significantly different from Xumai 33 (Figure 6). Under the same thousand-grain weight monitoring and the same VIs, although the three varieties generally showed the same negative correlation trend, the relative growth rate of Xumai 33 was significantly different from that of Xumai 44 and Xumai 43. Moreover, in the monitoring model construction of fresh thousand-grain weight, the best VIs of Xumai 33 were different from that of Xumai 44 and Xumai 43, which was the PSRI. The construction of monitoring models with different VIs also created growth rates in different directions. The monitoring models of the three varieties all maintained a roughly positive correlation in terms of water content. Nevertheless, the model of Xumai 33 exhibited general convex function behavior, and the models of Xumai 44 and Xumai 43 exhibited strong concave function behavior. The results showed that the fitting effect of the single-cultivar model was better, but the model differences were large.

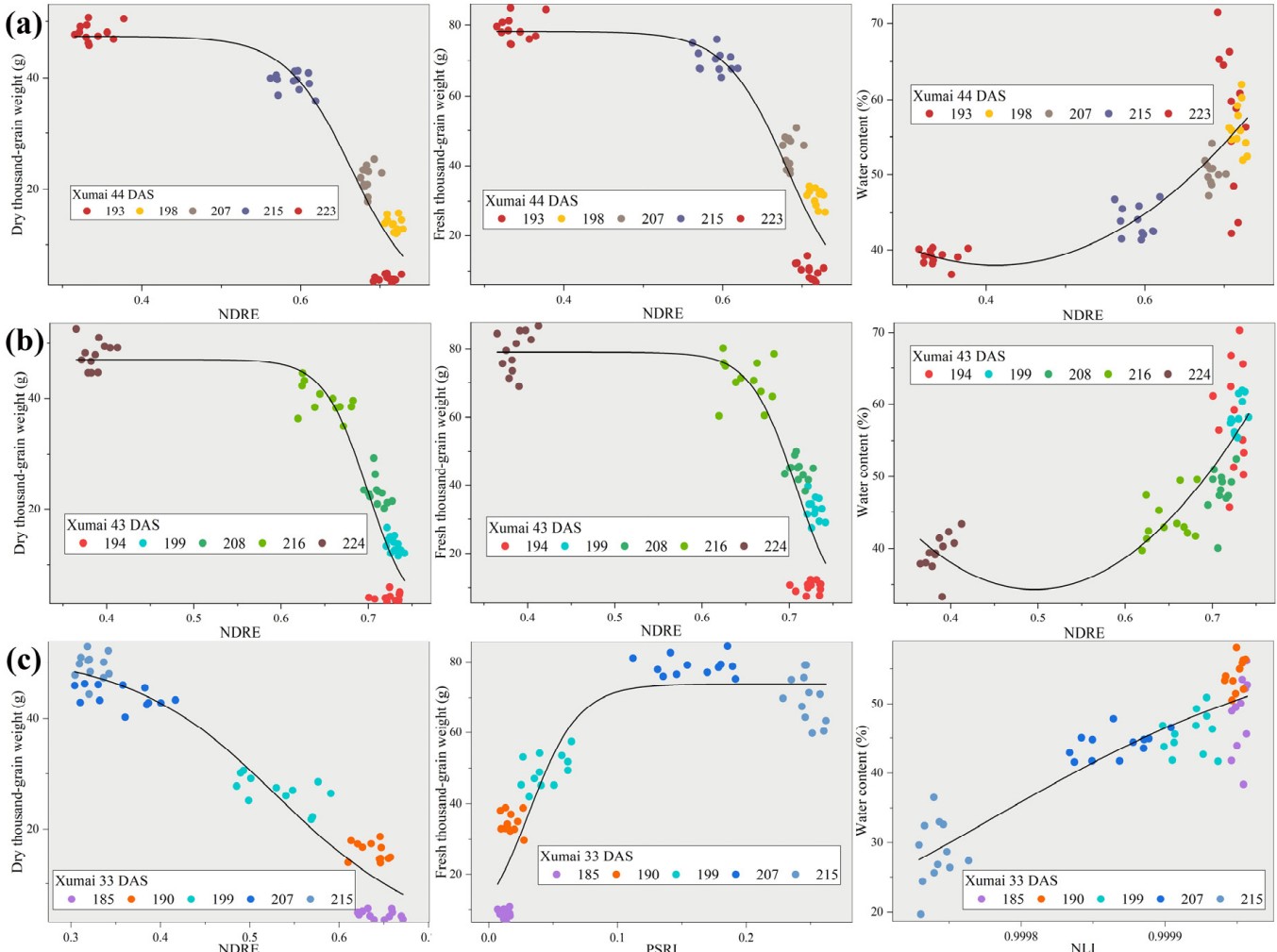

**Figure 6.** Monitoring models between three indicators of wheat grain-filling process and VIs during the growing season's last five periods in Xumai 44 (**a**), Xumai 43 (**b**) and Xumai 33 (**c**).

All samples were randomly divided into model development and model validation according to the ratio of 3:1. The number of validation samples for the overall model and the single-cultivar model was 60 and 20, respectively (Table 3). The 1:1 relationship diagrams between predicted values of established models and measured values were drawn to evaluate the accuracy of the monitoring models of grain-filling indicators (Figure 7).

Regardless of the overall model or the single-cultivar model, the monitoring effects of the three indicators of the wheat grain-filling process showed that the dry thousand-grain weight was the best, the second best was the fresh thousand-grain weight and the last was the water content.

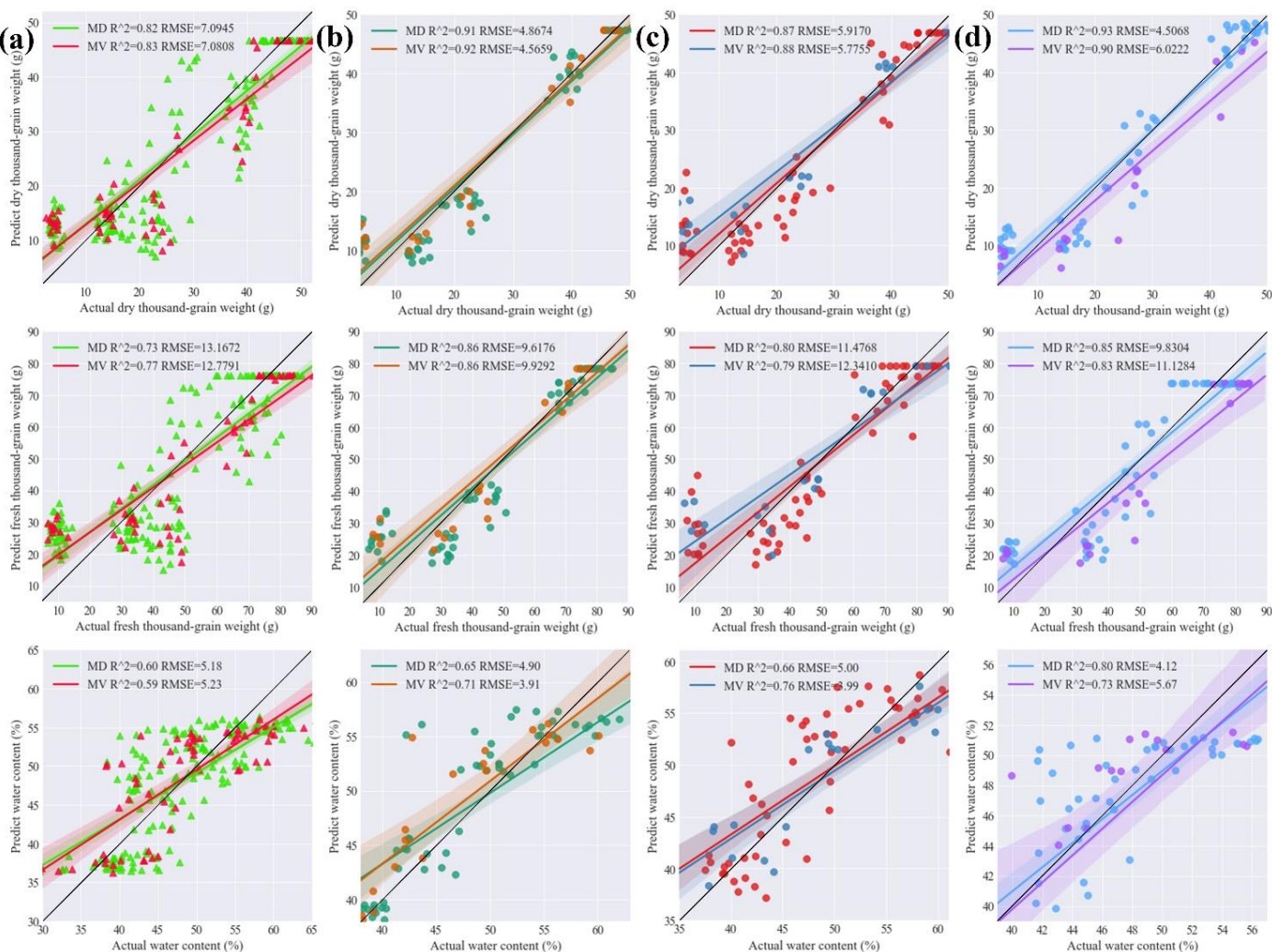

**Figure 7.** Reliability testing of the remote sensing monitoring model of three indicators of winter wheat in the grain-filling stage. (**a**) is the overall model, and (**b**–**d**) are the single-cultivar models of Xumai 44, Xumai 43 and Xumai 33, respectively.

In the continuous monitoring of the dry thousand-grain weight, the $R^2$ of all models on both the model development and the model validation reached more than 0.8. In the continuous monitoring of the fresh thousand-grain weight, only the $R^2$ performance of the single-cultivar model on the model development and model validation remained at around 0.8, and the effect of the overall model also remained at a good result with an $R^2$ greater than 0.7. The monitoring effect of water content basically remained at a good level of $R^2$ between 0.6 and 0.8. In conclusion, in terms of model validation of the three indicators, the effect of the overall model was slightly worse than that of the single-cultivar model, but the result of the overall model could basically satisfy the accurate description of the grain-filling process of winter wheat.

### 3.3. Thematic Maps of Winter Wheat Grain-Filling Process

Considering that the models of the three varieties are quite different, and the performance of the overall model can also satisfy the accurate simulation of the grain-filling process of the three varieties of winter wheat, this study used the overall model to visualize

the grain-filling process of winter wheat. The VIs have the potential to predict winter wheat maturity variability in commercial fields due to the strong correlation, high $R^2$, and low RMSE values observed. Figure 8 shows the different time inversions of dry and fresh thousand-grain weight inputted into the logistic growth model with the PSRI as the independent variable, and shows water content inputted into the polynomial model with the NDRE as the independent variable.

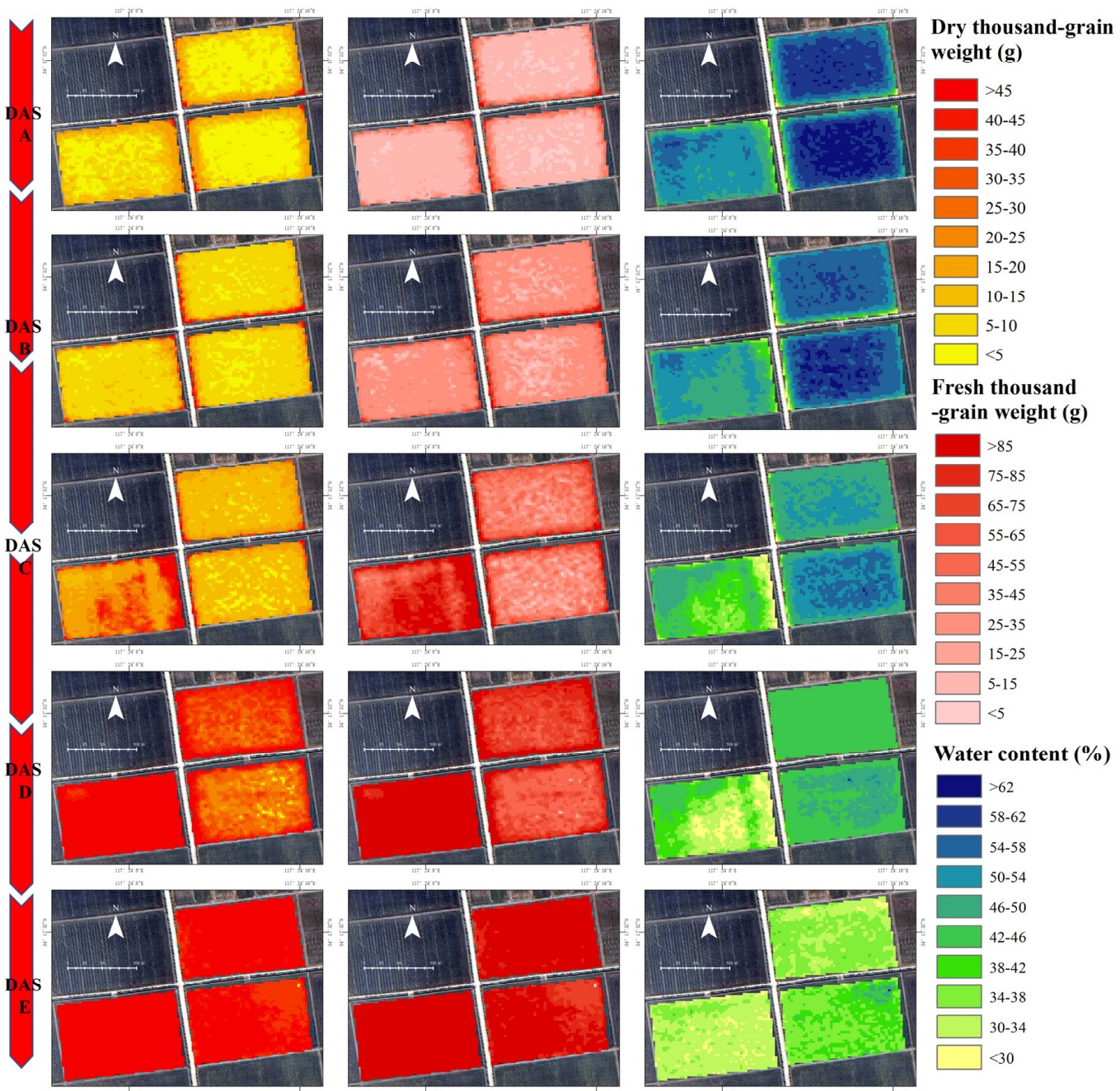

**Figure 8.** Spatial and temporal variability maps for three indicators of wheat grain-filling process using VIs for three fields.

Through time-series satellite imageries, there is a very intuitive visualization effect on the uniformity of the maturity of different wheat cultivars in the field environment. During the period from DAS A to DAS B, the wheat grain-filling process changed steadily; starting from DAS C, the grain-filling process accelerated and, finally, stabilized. Although the

sowing date of Xumai 33 is earlier than that of Xumai 44 and Xumai 43, it can be intuitively found from the remote sensing thematic map that the grain-filling process of Xumai 33 is earlier than that of Xumai 44 and Xumai 43, and its grain-filling rate is also faster than that of the others. The inversion of the grain-filling process of winter wheat in the time span based on VIs effectively achieves a breakthrough in the acquisition of point-to-plane information between and within cultivars.

## 4. Discussion

At present, the ability to rapidly monitor crop growth and yield over large scales based on remote sensing data is an area of active research. There are wide research studies and applications for field crop monitoring by remote sensing technology being conducted, such as on wheat, rice, soybean, maize and potato [18,43,48,49]. However, there are few reports describing the wheat grain-filling progress using time-series remote sensing data. The results of this study demonstrate the great potential of using time-series multispectral satellite imageries to accurately monitor the grain-filling process of multiple wheat cultivars.

Remote sensing has been widely exploited to predict yield and explain senescence in many crops, with senescence being the key point for identifying crop maturity. The changes in VIs also tend to be obvious when senescence increases [26,50]. In this study, the time-series Planet imageries covering the whole grain-filling process of winter wheat were used to explore its influence on the simulation of the grain-filling process by constructing VIs. As the best variable, the PSRI can accurately simulate the changing trend of dry and fresh thousand-grain weight in the overall model, and the NDRE has the best performance in multiple single-cultivar models (Table 3). The PSRI can be used to maximize the sensitivity of the ratio of carotenoids (such as α-carotene and β-carotene) to chlorophyll, and its increase predicts increased canopy stress, the onset of vegetation senescence and the maturation of plant grains [51]. However, this change in VI values may occur for a variety of reasons, affecting the final winter wheat yield. The uncertain grain-filling habit of wheat makes the determination of plant senescence points challenging. Despite that, the positive correlation between maturity and the PSRI found in this research showed that the VI values increased when winter wheat maturity increased (Figure 5). This presents a potential for the use of remote sensing to assist in the decisions made regarding harvest winter wheat fields.

It is obvious that the NDRE based on the red-edge band as an independent variable has a significant role in the wheat grain-filling process model. Similar to the NDVI, the NDRE is a vegetation index constructed based on the normalized ratio of NIR and red-edge bands to measure and analyze vegetation health in multispectral imageries. The red-edge band is the region between 680 and 750 nm and is considered the most significant sign of green vegetation. Because it has the point where the reflectance rises the fastest, this point is also known as the maximum value of the first derivative of the plant spectrum in this wavelength range [52]. As a transition band between the red and NIR bands, the red-edge band marks the boundary between chlorophyll absorption in the red region and scattering in the NIR region due to leaf internal structures [53]. Compared with the "saturation" phenomenon that the NDVI is prone to produce at the later stage of growth, the NDRE can reflect more sensitively the chlorophyll content of vegetation, such as after crop canopy closure [54,55]. In the early stage of crop grain formation, the demand for sugar molecules produced by photosynthesis is usually high, and the demand for sugar molecules is reduced near the harvest period. To a certain extent, chlorophyll reflects the ability of crop photosynthesis activities; therefore, using NDRE information can help us realize the acquisition of crop grain growth to optimize the harvest time based on transformation in photosynthesis.

Among the three indicators of the grain-filling process of winter wheat in this study, whether it is the overall model or the single-cultivar model, the simulation effect of the dry thousand-grain weight is the best, followed by the fresh thousand-grain weight, and the simulation effect of water content is the worst. With the continuous grain-filling process of winter wheat, the dry matter content of grains increased, whereas the relative water content

decreased. Through the comprehensive comparison of the monitoring models of the three indicators, it was found that the dry thousand-weight is the most ideal indicator to reflect the grain-filling process of winter wheat, with the reason being that the crop grain-filling process is the accumulation process of dry matter. However, the measurement of the fresh thousand-grain weight is greatly affected by the environment, such as temperature and humidity. The reason why the simulation effect of the fresh thousand-grain weight is worse than that of the dry thousand-grain weight in this study may be due to weather factors (precipitation, temperature and humidity) in different sampling periods, differences in the time period of sampling in a day, the duration of threshing at the laboratory and so on, which can lead to an error in the acquisition process of the measured data. Therefore, under the influence of these factors, the simulation effect of grain water content was worse than that of the dry and fresh thousand-grain weight.

The use of non-destructive methods, such as remote sensing, to predict winter wheat maturity could increase the management accuracy at harvest time, especially in countries or regions with more than one crop per year. For instance, winter wheat is used in a crop rotation with rice in most parts of Jiangsu Province, China, and this crop rotation system causes a tight time between the harvest of winter wheat and the planting of rice. The determination of the harvest point of winter wheat is very important in agriculture production, but generally, early harvest means lower yields, whereas delaying the harvest can imply more loss and damage, such as through the impact of the rainy season and lodging. Thus, the grower must identify the moment close to the ideal maturity of grains to start harvesting. It is known that, naturally, there is soil variability, among other environmental components, which leads to the uneven development of plants [25,56]. Hence, plants in more favorable locations will potentially be able to grow faster and mature earlier (Figure 8). Because the winter wheat growing around the field has the advantage of a marginal benefit, the overall growth and maturity are better than those in other locations, and the growers have limitations in judging the overall maturity of winter wheat in the field without an in-depth field investigation.

Overall, the results suggested that the application of time-series high-spatial/temporal-resolution satellite imagery can directly and practically help growers to identify which field is more developed and should be harvested first. The results provide directions and applications to improve winter wheat cultivation and production through better management and conduction of the crop. From a practical point of view, farmers, researchers and government agencies could use high-resolution satellite imageries to monitor crop growth and generate yield formation maps. This may be an interesting strategy to support decision-making in harvest and postharvest operations, such as deciding the optimal moment to start harvesting, when areas of the field should be harvested first, and product logistics and storage. In addition, it is possible to obtain the surface information of the uniformity of the maturity stage between varieties and within varieties based on an accurate description of the grain-filling process of winter wheat by high-resolution satellite imageries. This could be used to speed up plant breeding programs focused on increasing winter wheat population uniformity.

The dynamic monitoring model of the wheat grain-filling process for the mature period was constructed in this study and performed well in predicting the grain indicators. Some potential applications and extensions can be considered based on this conclusion, although more development in the following areas is still needed. Firstly, although the planting structure is relatively complex in Xuhuai District, Jiangsu, i.e., the preceding crops of winter wheat are diversified, the possibility of applying DAS-based dynamic monitoring models and results to other scenarios under specific regional climate conditions (Figure 2) is promising. Further research will focus on exploring the performance of models incorporating short-term and long-term climate variables in different soil environments and varieties.

Secondly, although there is great potential for PSRI and NDRE indices to predict winter wheat maturity, the use of remote sensing in association with other tools is suggested,

such as the cumulative growing degree-day (GDD). Several studies have shown that GDD models could be used successfully to monitor wheat growth and productivity under climate change [57,58]. For conditions in Jiangsu Province, China, all fields reached maturity around 2000 GDDs based on the impact of crop rotation. It implies that more research is needed to establish when the winter wheat cultivars reach maturity based on the GDD in Jiangsu Province, especially in a winter wheat breeding program, where breeders have been working to develop new short-cycle cultivars to be used with rice in crop rotation systems.

Thirdly, it is of great scientific significance that this approach, although developed in Xuhuai District, Jiangsu, can potentially be used in different regions, not only for winter wheat but also for other staple crops such as rice, soybeans and maize. The relatively simple nature of the model input data (optical remote sensing and crop growth stage) should make it convenient and easy to develop the application devices to generate these results. Imageries from satellite platforms are valid for the modeling, while combining smartphone applications and real-time imagery data acquisition could accelerate the application of such tools in practice.

### 5. Conclusions

This study clearly demonstrates the feasibility of using VI-based remote sensing information derived from Planet imageries to assess the grain-filling process of winter wheat. Three indicators of the grain-filling stage, i.e., dry thousand-grain weight, fresh thousand-grain weight and water content, had significant correlations with most VIs. The dry thousand-grain weight obtained the best inversion results in the overall model, and its $R^2$ exceeded 0.8. Wheat growers may be able to utilize satellite remote sensing to manage their areas remotely and invert the fields that entered the grain-filling period first, making the most accurate management decisions using spatial variability maps. However, future studies are needed to explore the efficiency of remote sensing in monitoring winter wheat maturity across more different cultivars and geographic locations, especially from an economic return perspective. In addition, the use of a thermal sum measure is recommended, such as growing degree days (GDDs), in combination with remote sensing, thus strengthening the potential to evaluate the monitoring of the grain-filling process considering the local environmental conditions.

**Author Contributions:** Conceptualization, X.Z. and Y.L. (Yangyang Li); formal analysis, X.Z. and Y.S. (Yawei Sun); investigation, X.Z., Y.L. (Yimeng Li) and Y.Y.; data curation, X.Z. and Y.S. (Yijun Su); writing—original draft preparation, X.Z.; writing—review and editing, Y.L (Yaju Liu). All authors have read and agreed to the published version of the manuscript.

**Funding:** This research was financially supported by the project financed through the research and development of plant variety testing technology, the Development Center of Science and Technology, MARA of the People's Republic of China.

**Conflicts of Interest:** The authors declare no conflict of interest. The funders had no role in the design of the study; in the collection, analyses, or interpretation of data; in the writing of the manuscript; or in the decision to publish the results.

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
