# Peer review of "Research on Dynamic Monitoring of Grain Filling Process of Winter Wheat from Time-Series Planet Imageries"

_agronomy, doi:10.3390/agronomy12102451_

Round 1

Reviewer 1 Report

Paper presents results of an experimental study that could be useful for potential readers interested in development of remote sensing tools for the analysis of the wheat grain filling process. Paper is well written and organized. It gives comprehensive overview of the conducted experiments (sample collection, analysis and model fitting), and reports major outcomes of the attempts to predict three wheat grain filling process indicators based on selected vegetation indices. Besides, it also investigates temporal correlations and relationships between conducted spectral measurements and three indicators, as well uncertainty of the fitted models. Analyses were done on different test sites, and for two different crop rotation scenarios. In general results speak for themselves, but it seems that there is no final solution for the tackled problem. In the discussion and conclusions this has been mentioned. It would be useful if there could be more explicit suggestion and statement whether this type of research could be easily transferable to other locations and scenarios, or it would require the same amount of effort in collection of field samples, their analyses and model fitting. I am aware that the answer is probably closer to the latter, which would require similar amount of work in order to apply the same methodology to other use case scenarios. However, it would be useful to have some discussion related to this, and what do you see as some possible ways to make these kind of tools more accessible and used in practice. Regarding the possibility to transfer the models and results of conducted analyses to other use case scenarios, the main interest would be to apply it to the same filed and crop variety in the next crop growing season, but there could also be different combinations, like the same region (soil and environmental conditions, with same or different crop species variety), or using the obtained measurements to make development of same tools in some other region. Ultimately, there is a question how the data corresponding to different crop rotation strategies could be combined or used, although the results in the paper suggest that there was significant difference between indicator variables for the field 33 with sweet potato crop rotation, in comparison to 43 and 44 with rice as the second crop. Please comment on this and add some advice what we could do to resolve these in the future (what type of research you see as the potential solution, since the widespread adoption will not be possible without such efforts).

______________________________

Please check caption of figure 6, word "actuall" is misspelled

Reviewer 2 Report

This paper is very interesting and well presented. I just have a few questions/comments and suggestions for consideration.

1. You used many vegetation indices in your model. I would assume many of them are highly correlated. How did you address the issue of multi-collinearity in your analysis? I would suggest adding a table or a correlation matrix to show the correlation between the VIs that you used and describe in a paragraph how you addressed the issue of multi-collinearity.

2. For Figure 7 I suggest you revers the legend to show lower values at the bottom and higher values on top of the bar to avoid confusion.

3. Move lines 322-348 to introduction. The background information provided in these paragraphs would best fit to the introduction. 

4. In lines 390-398 you mentioned the potential effects of weather factors on on fresh thousand grain weight. Would you consider adding short term and long term climate variable in your models in addition to the VIs?

I overall think this work is very valuable and fills out the gap of monitoring vegetation phenology using high spatial resolution remote sensing data. 
